# Developmental Methylome of the Medicinal Plant *Catharanthus roseus* Unravels the Tissue-Specific Control of the Monoterpene Indole Alkaloid Pathway by DNA Methylation

**DOI:** 10.3390/ijms21176028

**Published:** 2020-08-21

**Authors:** Thomas Dugé de Bernonville, Stéphane Maury, Alain Delaunay, Christian Daviaud, Cristian Chaparro, Jörg Tost, Sarah Ellen O’Connor, Vincent Courdavault

**Affiliations:** 1Faculté des Sciences et Techniques, Université de Tours, EA2106 Biomolécules et Biotechnologies Végétales, F-37200 Tours, France; thomas.duge@univ-tours.fr; 2INRA, EA1207 USC1328 Laboratoire de Biologie des Ligneux et des Grandes Cultures, Université d’Orléans, F-45067 Orléans, France; alain.delaunay@univ-orleans.fr; 3Laboratoire Epigénétique et Environnement, LEE, Centre National de Recherche en Génomique Humaine, Institut de Biologie François Jacob, F-92265 Evry, France; daviaud@cng.fr (C.D.); tost@cng.fr (J.T.); 4CNRS, IFREMER, UMR5244 Interactions Hôtes-Pathogènes-Environnments, Université de Montpellier, Université de Perpignan Via Domitia, F-66860 Perpignan, France; cristian.chaparro@univ-perp.fr; 5Max Planck Institute for Chemical Ecology, Department of Natural Product Biosynthesis, 07745 Jena, Germany; oconnor@ice.mpg.de

**Keywords:** alkaloids, DNA methylation, epigenetics, folivory, organs, plant specialized metabolism

## Abstract

*Catharanthus roseus* produces a wide spectrum of monoterpene indole alkaloids (MIAs). MIA biosynthesis requires a tightly coordinated pathway involving more than 30 enzymatic steps that are spatio-temporally and environmentally regulated so that some MIAs specifically accumulate in restricted plant parts. The first regulatory layer involves a complex network of transcription factors from the basic Helix Loop Helix (bHLH) or AP2 families. In the present manuscript, we investigated whether an additional epigenetic layer could control the organ-, developmental- and environmental-specificity of MIA accumulation. We used Whole-Genome Bisulfite Sequencing (WGBS) together with RNA-seq to identify differentially methylated and expressed genes among nine samples reflecting different plant organs and experimental conditions. Tissue specific gene expression was associated with specific methylation signatures depending on cytosine contexts and gene parts. Some genes encoding key enzymatic steps from the MIA pathway were found to be simultaneously differentially expressed and methylated in agreement with the corresponding MIA accumulation. In addition, we found that transcription factors were strikingly concerned by DNA methylation variations. Altogether, our integrative analysis supports an epigenetic regulation of specialized metabolisms in plants and more likely targeting transcription factors which in turn may control the expression of enzyme-encoding genes.

## 1. Introduction

The Madagascar Periwinkle (*Catharanthus roseus* (L.) G.Don) is a well-known producer of Monoterpene Indole Alkaloids (MIAs) [1]. Besides the use of some of its MIAs in anticancer chemotherapy, *C. roseus* has become an important model medicinal plant especially due to the availability of transcriptomic and genomic resources [2,3,4,5]. The common precursor to all MIAs is strictosidine which originates from the condensation of tryptamine with secologanin. Major end product MIAs in *C. roseus* are catharanthine and vindoline, which are able to dimerize into anhydrovinblastine, a molecule with antineoplastic properties. Interestingly, accumulation of MIAs in *C. roseus* is highly compartmentalized and displays organ specific patterns. While catharanthine is widespread in the plant, vindoline is restricted to the aerial parts [6]. Its synthesis is also more active in the basal (versus distal) regions of young immature leaves [7]. Vindoline results from the 7-step conversion of tabersonine [8], a molecule that is also produced in virtually all parts of the plant. However, tabersonine is specifically metabolized into vindoline in leaf or alternatively into lochnericine or hörhammericine in roots [9,10]. Tabersonine and catharanthine are converted from a common precursor by specific carboxylesterases named TABERSONINE and CATHARANTHINE SYNTHASE, respectively [11,12]. Adventitious roots obtained from cuttings incubated in water also display a root-like metabolic profile enriched with several tabersonine derivatives. Flowers have been reported to have a very low content in catharanthine and high amounts of serpentine and ajmalicine [13]. In addition to this tissue specificity, it has been shown that *C. roseus* subjected to herbivory by the Tobacco Hornworm *Manduca sexta* underwent an increase of MIA accumulation through a two-step process [14]. Folivory is thus accompanied by a fast and local accumulation of strictosidine and by a delayed systemic accumulation of terminal MIAs such as catharanthine and vindoline. Therefore, the MIA pathway in *C. roseus* appears to be specifically regulated at both developmental and environmental levels. However, the precise nature of the regulatory events remains largely unknown.

At the molecular level, the regulation of MIA biosynthesis is known to operate through major transcription factors. For instance, the upstream iridoid part of the pathway leading to secologanin production is upregulated by two basic Helix-Loop-Helix (bHLH) transcription factors named BIS1 and BIS2 though these transcription factors do not control the accumulation of later alkaloids such as catharanthine or tabersonine [15,16]. However, another bHLH transcription factor, CrMYC2, positively regulates MIA accumulation [17], through the concomitant regulation of the AP2/ERF ORCA transcription factors [18,19]. This illustrates how MIA biosynthesis is tightly coordinated by a complex regulation network [20]. Furthermore, it has recently been shown that vindoline accumulation in leaf is under the positive control of GATA transcription factors, which are in turn degraded by PHYTOCHROME INTERACTING FACTORS in darkness [21]. This complex regulatory network is also shaped by phytohormones that are known to modulate MIA accumulation. For example, it is well established that auxins downregulate the expression of several MIA related genes, while cytokinins and jasmonate promote MIA accumulation (reviewed in [22]).

Epigenetics refers to any non-genetic heritable molecular modification of the genome that may alter gene expression [23]. In addition and/or in combination with transcription factors, epigenetic modifications are known to shape and regulate many aspects of plant development and physiology DNA methylation at cytosine positions is particularly known to influence gene expression, transposon activity, and chromosome interactions [24]. It modulates plant development and response to environmental clues [24,25]. DNA cytosine methylation may occur de novo or by maintenance during replication in the CG, CHG, and CHH contexts (H = A, T or C), within or outside gene bodies and transposable elements (TE) [26]. The global DNA methylation level greatly varies among plant species [25]. However, general methylation in gene bodies appears to be a conserved process in orthologous genes from phylogenetically distant species [27] while methylation in promoters or first intron may control gene expression [24,28,29]. Plant development is controlled by specific DNA methylation patterns, e.g., during ontogenesis, but also in response to environmental stresses [29,30]. It has recently been showed that variation in DNA methylation patterns may affect gene expression in *cis* and in *trans* possibly via small RNAs affecting primary and specialized metabolic pathways in *Arabidopsis thaliana* (L.) [31]. In addition, histone and DNA modifications are likely to shape relationships between cell metabolites and the corresponding gene expression [32].

Specialized metabolic pathways in plants are developmentally and environmentally regulated and an additional epigenetic control has been proposed. For example, apple skin color in some cultivars is due to the accumulation of anthocyanins whose biosynthesis involves enzymes encoded by genes which are differentially methylated at cytosine nucleotides [33]. In addition, hypomethylation of flavonoid related genes has been correlated with increased expression and accumulation of anthocyanins. In non-model plants used in traditional medicine, DNA cytosine methylation has been shown to be an important factor controlling the accumulation of specialized metabolites during in vitro growth. For example, variations in methylation profiles have been shown to vary between wild and cultivated ginseng and were associated with differential specialized metabolite accumulation [34]. In addition, inhibitors of DNA methylation are able to increase phenolic product biosynthesis in *Salvia miltiorrhiza* (Bunge) hairy root cultures [35]. Finally, a correlation was made between methylation state and Benzoisoquinoline Indole Alkaloids in different organs and cultivars of opium poppy [36]. However, an integrative analysis with multi-omics data is still lacking to establish and precise the role of DNA methylation in the regulation of specialized pathways in medicinal plants.

In the present study, we investigated whether variations in DNA methylation profiles may control the developmental and environmental regulation of the MIA biosynthetic pathway in *C. roseus*. We analyzed genome-wide DNA methylation comprehensively at single nucleotide resolution by Whole Genome Bisulfite Sequencing (WGBS) focusing on either the entire genome or regions inside/around genes under 9 different conditions (different organs or parts, during their maturation and in response to biotic attack) and compared gene associated methylation profiles to the respective gene expression levels measured by RNA-seq and metabolomic MIA accumulation.

## 2. Results and Discussion

### 2.1. C. roseus Tissue-Specific Methylome Is Characterized by Context-Dependent Variations within Genes and Their Flanking Regions

Genome-wide methylation profiles may be measured by several techniques relying on high-throughput DNA sequencing methods [37], of which WGBS provides the most comprehensive coverage and can be considered as the current Gold standard technology. Genome-wide cytosine methylation in *C. roseus* was analyzed using WGBS in organs and conditions reflecting the most pronounced contrasts in qualitative and quantitative MIA accumulation (Figure 1A). The objective was to highlight relationships between gene methylation and expression in main organs such Root, Flower (Flow), Young Leaf (Yleaf), Mature Leaf (Mleaf), and adventitious Roots [10] obtained from stem cuttings (Aroot). Since MIA biosynthesis has been reported to be more active in basal parts of young leaves [7], we separately analyzed basal (Basal) and distal (Distal) parts of young leaves. Due to the important stimulation of the MIA pathway observed in plants subjected to herbivory [14], we also compared two samples from fed plants. The first one corresponds to the intact part of leaves damaged by caterpillars and sampled 24 h after the feeding process (Mleafcat). The second sample corresponds to intact newly emerged leaves collected one week later after an initial attack on older leaves (Yleafcat). The entire genomic sequence of *C. roseus* as well as a reduced representation on gene models with +/− 5 Kbp around were used to map reads (Figure 1B).

Mapped reads ranged from 33 to 192 millions on the entire genome (about 0.5 Gb; [3]; Appendix A; see the Material and Methods Section for conditions of WGBS analysis). The total number of covered C sites varied but a total of 637,916 sites (520,629 CHH, 63,702 CHG and 53,585 CG) out of 37,315,277 were covered by at least seven reads in all nine samples. Decreasing DNA methylation levels were observed between the CG, CHG, and CHH contexts (Figure 1B) as in many plants [29,38]. Mapping against the gene models followed a similar trend to that observed when focusing on the entire genome but with lower methylation levels and similar methylation percentages for CG and CHG contexts (Figure 1B). The decreased percentage of methylation observed within the gene models (Figure 1B) suggested that intergenic regions were strongly methylated notably in CG context probably in relation to the hypermethylation of TE sequences. This global methylation profile of *C. roseus* showed little variation among tissues, which is in agreement with previous observations for other plants [29,38], and strongly similar to apple trees [39] and soybean [40]. This suggests no clear relationship with plant phylogenetic or genome size with methylation patterns, while it has been reported that both DNA methylation (CG and CHG mainly) and TE abundance are positively correlated with genome sizes in plants [27,29,38,41].

Methylation in genes showed a specific depletion in both 5’ and 3’ UTRs in the 3 cytosine contexts (Figure 1C). In addition, intergenic regions together with upstream and downstream sequences had a significantly higher number of Differentially Methylated Cytosines (DMC; pair samples comparison for each cytosine) in the CHG and CHH contexts (Wilcoxon rank sum test, *p* < 1 × 10${}^{-5}$) in contrast to the other features related to gene body (Figure 1D). Contrastingly, gene body methylation (exon, intron and 3’UTR) displayed more DMCs in the CG context than in the two other contexts (*p* < 0.001). This observation suggests a role of gene body methylation in promoting organ differentiation, as it has been found in *Brachypodium dystachion* (L.) [42]. However, we observed a depletion of DMCs in the CG context within 5’ UTR regions (*p* < 0.05; Figure 1D) in addition to the global methylation depletion (Figure 1C). Altogether, these data showed that tissue-specific variations of DNA methylation in/around gene models (in terms of DMC numbers) are context and sequence dependent, with high variations in CG for gene body and in CHG/CHH for regulatory sequences (Up and Downstream), and low levels and variation in the 5’UTR (proximal promoter). While there is emerging evidence that differential methylation plays a role in tissue specific gene expression in plants [17,40,43,44], only few studies have measured methylation at single-base resolution for both DMC and DMR analysis [42]. This last report analyzed variation of DNA methylation in two tissues (leaf and floral bud) in the model grass species *B. distachyon*. They showed that tissues are similar and most of the observed differences (0.5 to 2.5% of DMC/C sites) occurred in the CHG context within TEs and promoters compared to genes. Here, we confirm in seven different tissue types weak global variations in DNA methylation but with an increased range of affected sites (5 to 13% DMC/C sites). We also highlight that these variations are significantly affecting sequences within or flanking genes in a context-dependent way. For example, we found that promoter regions composed of both 5’UTR and Upstream sequences displayed opposite patterns of methylation and variations in a tissue-specific context analysis.

Focusing on reads mapped onto gene models (+/− 5 Kpb), we further analyzed the differentially methylated cytosines (DMCs) and regions (DMRs) in the different comparisons (Figure 2A). In all comparisons considered among the nine samples, many more DMCs were found in the CG (16,024) and CHG (17,432) contexts than in the CHH context (4667). This difference was however less pronounced at the DMR level (CG: 4374; CHG: 4508; CHH: 2604). Detection of differential methylation at the DMR level has the advantage of integrating several single loci and is probably more robust as reducing stochastic DNA methylation variation and better reflecting the overall DNA methylation status of a genomic region. Here, DMRs in the CG and CHG contexts seemed to encompass many DMCs, as suggested by the difference in DMC and DMR numbers. Large variations were observed in the number of DMCs or DMRs depending on the biological comparison considered. For example, a total of 3202 DMCs were identified between Root and Yleafcat, while only 95 were detected between Distal and Yleaf.

Three clusters of comparisons clearly emerged from the number of DMCs/DMRs (red boxes in Figure 2A): a first group containing the Yleafcat sample displaying many DMCs/DMRs in the CG and CHG context (bottom part of the tree in Figure 2A), a second containing Flow, Yleaf, and Mleaf with fewer DMCs/DMRs and a third group (top part of the tree) with a higher rate of differential methylation in the CHH context. The comparisons could indeed be efficiently separated according to their number of DMC/DMR by a principal component analysis (PCA, Figure 2B). The strongest differences concerned the number of DMCs in the CG (notably DMCs CG in introns) and to a lower extent CHG contexts (highest contribution of the corresponding variable to dimension 1 which represented 52% of the total variance) and clearly separated comparisons involving Root or Yleafcat samples (on the right part of the plot) to the others (on the left part). Dimension 2 represented less variance (12%), and was correlated with the number of DMRs in the CHH context in exons. It thus separated comparisons such as Flow vs. Yleafcat from Root vs. Mleaf for example. Interestingly, this analysis revealed specific patterns in differential methylation across samples, which could be further classified according to their methylation profiles (Figure 2C,D). Indeed, the hierarchical clustering obtained from the number of DMCs/DMRs differed according to the context considered. In contrast, the hierarchical clustering obtained from differentially expressed genes (DEGs) (Appendix A) strongly segregated the five main tissues (Root, Adventitious root, Flower, Mature, and Young leaves). The DMR clustering in CG, CHH and to a lower extent CHG separate organs (Root, Adventitious root, Flower, and leaves samples), although less pronounced, but also one condition (Yleafcat). However, we found a constant group containing Mleaf, Mleafcat, Yleaf, and Distal samples. Fewer DMC/DMR were detected among these samples (Figure 2A).

From a biological point-of-view, these observations indicate that minor changes in cytosine methylation status occurred between Mleaf and Yleaf but also in the mature leaves challenged by caterpillars, Mleafcat vs. Mleaf. By contrast, Aroot, Yleafcat, Root, and Basal samples generally formed separate groups in the hierarchical clustering. These samples were generally associated with more DMCs/DMRs, suggesting they are associated with particular methylation states that may reflect their organ function or response to biotic stress. Indeed, while all samples exhibited similar numbers of hypo or hyper methylated DMCs or DMRs, a large number of hypomethylated DMCs and DMRs were detected when comparing the Yleafcat sample to the other samples (Figure 2E). This specificity can explain the clustering of Yleafcat in our analysis (Figure 2C, D) but also supports a recent report about the depletion in the folate content that decreases global DNA methylation levels, while promoting plant resistance to a bacterial pathogen [45,46]. Thus, folate plays an essential role in the biosynthesis of a precursor of S-adenosylmethionine, the primary methyl group donor for most biological methylations [45,46]. Whether a particular depletion in folate content occurred in Yleafcat remains to be determined, it might be related to the activation of a systemic defense response due to an early folivory event.

Altogether, using 36 pairwise comparisons, our data support the idea that tissue-specific variations in methylation do not occur at random but are rather context-dependent and target specific parts of gene and flanking sequences. In addition, our integrative analysis (Figure 2) using PCA and clustering unraveled the significance of CG methylation for tissue-specificity, and CHG and CHH to a lower extent, besides questioning relationships with differential gene expression (Appendix A).

### 2.2. Tissue-Specific Covariations between Context-Dependent DNA Methylation and Differential Gene Expression May Affect Cellular and Physiological Functions

We next evaluated whether differential DNA methylation correlated with gene expression (Figure 3; Appendix A). A total of 18,839 genes were differentially expressed in at least one comparison (Figure 3A). All comparisons combined, differential methylation concerned 4866 and 3234 gene models at the DMC and DMR levels, respectively. The ratio between the number of genes that were only differentially methylated (DM) and those being simultaneously differentially expressed and methylated (thereafter named DEM) varied across comparisons. Firstly, it was neither linked to the total number of DEGs nor to the total number of DMCs/DMRs (Figure 3A). For example, the highest number of DEGs was found in comparisons between roots and aerial organs, but these comparisons did not display the largest number of DMCs/DMRs. Among gene models displaying DMCs, 22%, 16%, and 28% in the CG, CHG and CHH context, respectively, were also differentially expressed. The lower value in the CHG context was significant (Student’s *t*-test, *p* < 0.01) and indicated a lower overlap between differential expression and methylation in this context at the DMC level. Similarly, 28%, 21%, and 27% of the genes harboring DMRs in the CG, CHG, and CHH context, respectively, were also differentially expressed. Such ratios are in accordance with those found in other plants such as *Brachypodium* [42] where differential CG methylation explained between 1% and 9% of variation of tissue-specific variation in gene expression, soybean [40], or poplar [47]. By contrast, in rice subjected to cadmium stress, a larger overlap between DEG and DMCs/DMRs (>50%) has been detected in aerial parts of the plants [48].

We found that DNA methylation profiles of DEMs differed slightly from that of DM genes (Figure 3B; Appendix A). It appeared that DNA methylation was higher in differentially expressed genes. Indeed, the absolute difference of averaged methylation levels between DEMs and DM genes significantly differed from 0 in almost all contexts and gene features except UTRs (Wilcoxon rank sum test), while differences among contexts or among features were not significant. This may indicate that methylation of DEM genes may potentially occur in all features and that methylation levels may rather influence gene expression. This observation is somewhat similar to a recent study which used machine learning based algorithms to predict whether a gene is differentially expressed according to its methylation profile [49].

Overall, the relationships between gene expression and methylation level are complex [24,28,50,51]. Correlation may switch from positive to negative values according to environmental variations such as recently described in palm roots exposed to high salinity [52]. In our dataset, we found a significant and negative correlation between methylation in the CG context and gene expression at both DMC and DMR levels (Figure 3C,D; see, for example, CG in introns, Appendix A. A similar observation has been made in other plant species such as in tea [53]. DMCs/DMRs in the CHH context were globally less frequent in gene bodies and no clear cut relationship emerged between expression and methylation in the upstream or downstream regions. The strong positive correlations observed in Figure 3C within the CHH context are due to few occurrences (e.g., 2 points in exon, 4 in introns). The situation was less contrasted in the upstream and downstream regions of gene models, with more genes showing a positive relationship between methylation and expression (Appendix A). Whether these observations go beyond a simple correlation remains a matter of debate [54].

Within gene bodies, methylation was more frequently observed in introns and exons than in UTRs and inverse correlations between methylation level and logFC of expression clearly appeared. Although the role of gene body methylation still remains unclear [54], a recent report suggested it may maintain gene expression consistent among cells in *A. thaliana* roots [55]. Concerning tissue-specificity, our data are in agreement with previous reports showing that CG methylation covaries with differential gene expression between tissues explaining up to 9% of the variation [42]. Similarly, tissue-specific gene expression patterns have been found to be negatively correlated with both gene body and promoter methylation in poplar [44].

We next focused on DMRs and averaged them per gene loci to identify genes that were simultaneously differentially expressed and methylated in the same comparison. According to methylation differences between samples, we were able to detect 12 groups (from K1 to K12) using a correlation-based clustering with a dynamic tree cut [56] (Figure 4A). These groups contained specific gene sets with specific DNA methylation and expression profiles (Figure 4B). Gene functions were functionally analyzed by mapping genes to the Gene Ontology (GO) for each comparison separately (Appendix A). Functions for each of the 12 groups were also analyzed by GO enrichment analysis (Appendix A) as well as by using eggNOG mapping (Figure 4C). The lists of DEMs per cluster and per comparison are given in Appendix A, respectively, and summarized using their most represented UNIPROT keywords regarding their levels of methylation and expression (Figure 4B). The following sections describe the most relevant biological process where DEMs were identified in the different comparisons.

#### 2.2.1. Photosynthesis

Functional annotation with eggNOG revealed that many genes related to Energy production and conversion (group C, Figure 4C) were differentially methylated in several groups (K2, K6, K7 and K9 especially). Behind this wide-scale annotation, functions related to photosynthesis were found. Our dataset indeed suggests a strong control of the photosynthesis process by DNA methylation. GO term enrichment (Appendix A) clearly revealed a significant over-representation of genes related to photosynthesis in comparisons involving roots and flowers. We found a good correlation between hypomethylation and increased expression for these genes (Figure 4B and Appendix A). It is therefore likely that the establishment of photosynthesis in green tissues may be triggered by a strong hypomethylation of photosynthesis-related genes. Such an epigenetic control of nuclear photosynthesis-related genes has already been observed in Sycamore (*Acer pseudoplatanus* (L.)) suspension cells [57] and in Chinese white Poplar (*Populus x tomentosa* (Carrière)) [58]. A recent study also observed a strong correlation between hypomethylation of photosynthesis-related genes and over expression of the corresponding gene in young apple seedlings [59]. Hence, control of this physiological process by DNA cytosine methylation could be a conserved regulatory mechanism. Chloroplast related genes were downregulated while being hypermethylated in roots, flowers, and adventitious roots (Figure 4D). Contrastingly, these genes showed higher expression levels in green tissues accompanied by an hypomethylated state. In fact, mature leaves were expectedly enriched with photosynthesis related genes that were hypomethylated in this tissue. Caterpillar-challenged mature leaves had a slightly modified profile in contrast to mature leaves, with less genes related to photosynthesis. This may be explained by the general reduction in photosynthesis during herbivory or response to wound mediated by the Jasmonic Acid signaling pathway [14,60].

#### 2.2.2. Defense Response

It is well-established that salicylic acid treatments or challenges with phytopathogens trigger strong modification of the DNA methylation state [61,62]. In our dataset, the GO term “defense response” was found to be significantly enriched in several comparisons (Appendix A). For instance, a large reprogramming in the plant defense system characterized the Yleafcat sample (Figure 4D). These leaves were collected from plants one week after having been fed by caterpillars. In this organ, an important increase of MIA has previously been found in contrast to control leaves, suggesting a systemic activation of the plant defense system. DEMs genes in this Yleafcat sample particularly corresponded to transcription factors and kinases (Figure 4D and Appendix A). Interestingly, we found an ortholog (*CRO_T110712*) of the *A. thaliana* flavin-containing monooxygenase FMO1 (At1g19250), which is known to regulate systemic acquired resistance in distal organs [63]. Its ortholog in *C. roseus* was significantly higher expressed and hypomethylated in the Yleafcat vs. Distal comparison. In addition, several defense-related genes were overexpressed and hypermethylated in roots, including two LRR encoding genes, *CRO_T111352* and *CRO_T110750*, and a transcriptional activator of Pathogenesis-related gene transcriptional activator *CRO_T108223* (Appendix A).

#### 2.2.3. Cell Signaling

Several significantly enriched GO terms were related to signaling and regulation of transcription (Appendix A). For example, genes encoding transcription factors such as DIVARICATA (*CRO_T102972*) and TCP8 (*CRO_T111367*) were hypomethylated and higher expressed in roots relative to aerial organs (leaf distal part, flower, and mature leaf). A similar pattern was found for an ortholog of *A. thaliana* BHLH32 (AIG1) (*CRO_T104017*) known to be involved in root initiation [64]. Two auxin related genes overexpressed in roots were found (an Auxin efflux carrier component 1, *CRO_T110218*, and a Serine/threonine-protein kinase D6PKL2, *CRO_T110370*), as well as a cytokinin response regulator (ARR6, *CRO_T102339*) and an ethylene response factor (CRF4, *CRO_T101572*). Homologs of transcription factors such as ethylene related AtRAV1 or zinc finger ZAT9 were overexpressed and hypomethylated in adventitious roots. Similar modifications were also found for an UGT74E2 ortholog (*CRO_T110662*), which is known to regulate *A. thaliana* architecture by modifying auxin fluxes [65]. Such modifications in the DNA methylation levels may control the global auxin signaling pathway [66].

#### 2.2.4. Specialized Metabolism

Several genes related to secondary metabolism were identified to be more expressed and hypomethylated in adventitious roots (groups K4 and K11, Figure 4). A CYP72-encoding gene (*CRO_T103884*) potentially related to the secoiridoid metabolism was particularly overexpressed in adventitious roots vs. distal leaf parts. Two potentially MIA related genes were overexpressed and hypermethylated in roots in contrast to green tissues, an acetyltransferase (*CRO_T108440*) and an ethylene response factor named ORCA2 (*CRO_T110365*). Some specialized metabolism-related genes were overexpressed and hypomethylated in roots in contrast to basal leaf parts, as we found for a *CYP72* (*CRO_T106493*), a *CROCETIN GLUCOSYLTRANSFERASE* (*CRO_T108522*), and a *7-DEOXYLOGANETIN GLUCOSYLTRANSFERASE LIKE* (*CRO_T106923*).

Altogether, these functional analyses demonstrate that differentially methylated and expressed genes can be clustered into 12 functional groups reflecting broad processes such as stress response, photosynthesis, hormone signaling, and specialized metabolism. These clusters are in good agreement with the nature and characteristics of each sample and may participate in their cellular and physiological tissue-specificity.

### 2.3. Control of the Monoterpene Indole Alkaloid (MIA) Pathway Involves Variations in DNA Methylation in Genes Encoding Enzymes and Transcription Factors

*C. roseus* accumulates many MIAs following an organ- and tissue-specific distribution (Figure 5A,B). Results from previous LC-MS analyses of alkaloid content revealed for instance that major MIAs in aerial parts are vindoline and catharanthine. Tabersonine derivatives such as lochnericine are more specific to roots. Expression profiles of MIA related genes correlate well with this spatial separation as illustrated by T19H (encoding the TABERSONINE 19 HYDROXYLASE) and TAT (encoding the 19-HYDROXYTABERSONINE ACETYL-TRANSFERASE) which were clearly more expressed in roots and adventitious roots (Figure 5C). Genes involved in the conversion of tabersonine to vindoline (T16H2, 16OMT, T3O, T3R, NMT, D4H and DAT) were more strongly expressed in green organs, but not in flowers in agreement with vindoline accumulation [6]. Some genes were also strongly induced in Mleafcat, such as LAMT (encoding the LOGANIC ACID METHYLTRANSFERASE) and SGD (encoding the STRICTOSIDINE GLUCOSIDASE). Due to their contrasted expression, we focused specifically on the methylation status of MIA genes to determine whether DNA methylation might participate in regulating their expression levels.

MIA genes clearly display specific expression profiles and many were differentially expressed in the conditions investigated here (Figure 5C). We found 10 MIA genes out of 55 that were simultaneously differentially methylated and expressed (Figure 5C) and distributed along the MIA pathway (Figure 5A). Most of the differentially methylated loci were found in the upstream or downstream parts of gene models (Appendix A). Eight MIA genes contained DMCs (*8HGO, SLS2, T16H1, T16H2,* and *NMT*), as well as the three transcription factors (*BIS2, ORCA2,* and *ORCA3*) and four MIA genes contained DMRs (a strictosidine transporter *NPF2.9, TS, NMT* and *ORCA2*). Correlations between methylation and expression levels were globally negative (DMC: Pearson’s correlation coefficient −0.55, *p* = 0.002; DMR –0.95, *p* = 0.0002), suggesting a strong negative impact of DNA methylation on MIA gene expression. BIS2 encoding gene was hypermethylated in the CHG context in roots relative to Basal and YLeacat, while showing high expression levels in these two samples. *T16H2* followed a similar pattern in roots (hypermethylated, less expressed) relative to flower, Distal, and Mleaf.

Among the DEM genes, the *TABERSONINE SYNTHASE* (*CRO_110304* = *TS*) was more strongly expressed in young leaves from previously attacked plants (YLeafcat) in contrast to mature leaves (MLea). This gene belonged to group K2 (Figure 4) and was hypomethylated in Yleafcat in the downstream part (Appendix A). This is in accordance with the higher accumulation of vindoline in young leaves collected from previously challenged plants (Figure 5B). Indeed, the lower tabersonine accumulation in these tissues may be related to the increased expression of genes from the vindoline pathway and the resulting conversion of tabersonine into vindoline (Figure 5C). The gene encoding the transcription factor ORCA2 displayed two DMRs in the upstream and downstream regions, respectively (Appendix A). The hypermethylation in these regions corresponded to a higher expression in roots vs. mature leaves (Mleaf), attacked mature leaves (Mleafcat), and young leaves from previously fed plants (Yleafcat). *ORCA2* was indeed very weakly expressed in these green tissues. On the other hand, *NMT* was hypomethylated and more strongly expressed in young leaves from fed plants in contrast to roots. Because vindoline biosynthesis is inactive in roots and stimulated in Yleafcat (Figure 5B), its hypomethylation could be a mechanism to increase expression of vindoline biosynthetic pathway genes during the establishment of a systemic defense response.

In comparison with the hypomethylated state of flavonoid biosynthetic genes increasing both their expression and anthocyanin accumulation in apple fruit skin [33], methylation of MIA genes may appear as limited. A recent study has reported an inverse correlation between a P450 encoding gene methylation level and its expression [67]. Expression was induced upon Jasmonic Acid elicitation in seedlings of *Platycodon grandiflorus* and related to a higher accumulation of the triterpenoid saponin platycoside. Again, this was unlikely the case for the MIA in *C. roseus*. We may however envision that central transcription factors may be epigenetically regulated and which in turn control the expression of multiple downstream genes. Accordingly, we found three MIA-related transcription factors among our DEMs (Figure 5). In addition, we investigated whether other transcription factors in our DEMs may be co-expressed with MIA genes. Pairwise Spearman correlations between MIA and all DEM genes were calculated and relationships with a Spearman’s *rho* > 0.95 were kept for final network construction and visualization (Figure 6; Table 1).

MIA genes formed at least three tightly connected groups. A first one (community 1) containing the vindoline-related genes *T16H2, 16OMT, T3R, NMT, D4H,* and *DAT*. Co-expressed to these genes were also found many genes related to the seco-iridoid branch (*GES, IS, IO,* and *DL7GT*). The second one (community 2) contained genes related to terpene precursors (*DXS1, DXS2, HDS, HDR,* and *MECS*), seco-iridoids (*G10H* and *8HGO*). The third main group contained genes encoding strictosidine aglycone processing (*GS1, GS2, THAS1, THAS3,* and *THAS4*). A total of 24 transcription factors were found in the co-expression network (Table 1). In the first group (community 1, Appendix A), four transcription factors were found to be co-expressed with the MIA genes indicated above: a regulator of phloem development (*CRO_T102172*), a bHLH (*CRO_T109632*), a positive regulator of gibberellin action (*CRO_T108449*), and a WUSCHEL-related homeobox protein encoding gene (*CRO_T110203*). We also found three transcription factors co-expressed with the second main group: 2 bHLH (*CRO_T107249* and *CRO_T110065*) associated with pollen maturation in *A. thaliana* and a Myb-related protein (*CRO_T107392*). These two groups were tightly connected with the regulation of developmental processes. This suggests a strong coordination between MIA biosynthesis and the plant developmental program, as it has been previously observed [68]. Only one transcription factor was co-expressed with the third community containing genes encoding enzymes using strictosidine aglycones as substrate, a bHLH transcription factor (*CRO_T110954*). In addition to these three communities, we found that *ORCA3* and *STR* were strongly co-expressed (Figure 6). This was expected because it has previously been shown that MYC2 promotes *STR* expression through ORCA2 and ORCA3 [17,69]. In this small community, two other transcriptional activators were detected, a bHLH (*CRO_T101111*), and a Pathogenesis-Related gene transcriptional activator (*CRO_T108223*). Similarly to ORCAs, the latter also binds to the GCC box of defense related genes. Among the four transcription factors co-expressed with *SLS2*, and *IDI1*, two genes encoded bHLH (*CRO_T102366* and *CRO_T110248*). Interestingly, *CRO_T110248* is an ortholog of *A. thaliana ICE1* which codes a factor promoting stomatal differentiation in epidermis [70]. Because enzymatic steps in the MIA pathway are located in leaf epidermis after the DL7GT step, it might be relevant that late steps leading to strictosidine (LAMT, SLS, and STR) may be also developmentally regulated.

Altogether, our data show that a substantial number (10 out 55) of the MIA genes are differentially methylated and expressed. These genes exhibited strong correlation between DNA methylation and expression changes in agreement with the tissue-specific MIA accumulations. They also include critical enzymes of the pathway described as bottlenecks for MIA synthesis including 8HGO, SLS, and T16H [6,71,72]. This is also the case for NPF2.9 which controls strictosidine export from the vacuole [73]. Strictosidine is the precursor of all downstream MIAs and controlling its subcellular distribution could help to control the pathway activity. Biosynthesis of catharanthine and tabersonine only differs in the last step which is catalyzed either by the CATHARANTINE SYNTHASE (CS) or the TABERSONINE SYNTHASE (TS) [11,74]. Hence, a tight regulation of this last step may help to redirect metabolic fluxes towards either catharanthine or tabersonine and its associated derivatives such as vindoline. The hypomethylation and overexpression of TS in young leaves from previously fed plants could thus be an important regulation mechanism to control such fluxes as a plant defense response. TS has been shown to interact more strongly with the previous enzymes DPAS (DEHYDROPRECONDYLOCARPINE ACETATE SYNTHASE) than CS [11], reinforcing its crucial and possibly regulatory role in this bifurcation. In addition to this direct regulation of expression of biosynthesis-related genes by differential methylation, we found the concomitant differential methylation and expression of three major transcription factors controlling the MIA pathway. This suggests that DNA methylation may modulate MIA pathway activity by acting on upstream regulatory components. Complementary to this, 24 co-expressed transcription factors were among DEM genes suggesting a possible involvement in controlling the tissue-specificity of the MIA pathway.

## 3. Materials and Methods

### 3.1. Plant Material

*Catharanthus roseus* plants (cv. Apricot Sunstorm) were grown in a greenhouse under artificial light with a photoperiod of 16 h/8 h at 28 ${}^{{}^{\circ}}$C/23 ${}^{{}^{\circ}}$C. Different plant parts were collected (Figure 1; Table 2) and snap-frozen in liquid nitrogen and stored at −80 ${}^{{}^{\circ}}$C before processing. When possible, all samples were taken from the same plants and at least 10 biological replicates were prepared from ca. 3 month-old plantlets. Roots (Root) were quickly washed with sterile ultrapure water and dried before freezing. Old root parts (showing a brown aspect and strongly lignified) were discarded. Flower buds (Flow) were collected together with their peduncles. Mature (Mleaf) and young leaves (Yleaf) corresponded to the third and first fully expanded leaves respectively. Basal (Basal) and distal (Distal) parts were obtained by splitting young leaves in half. Herbivory treatments were performed as previously described [14]. *Manduca sexta* caterpillars (L3 stage) were dropped on third expanded leaves and allowed to feed for three hours and subsequently removed. Plants were kept if at least 25% of the attacked leaf was removed by the caterpillars. On one hand, damaged mature leaves were collected 24 h after the attack (Mleafcat). On the other hand, plants were maintained in the greenhouse and the first fully expanded leaf was collected one week after (Yleafcat). Adventitious roots were obtained from cuttings as described in [10]. Briefly, shoots were cut from their roots and maintained in tap water until white and soft adventitious roots appeared (usually within one month).

### 3.2. Whole Genome Bisulfite Sequencing (WGBS)

DNA was extracted from at least 10 individuals per sample type using the modified Trizol (Life Technologies) procedure involving the Back Extraction Buffer according to the manufacturer’s recommendations. The molarity of the samples was assessed by qPCR using the KAPA Library Quantification Kit from Roche and the size of samples was determined with the LabChip GX Touch Nucleic Acid Analyzer and the HT DNA 1K/12K/High sensitivity chip from PerkinElmer. Samples were pooled in equimolar conditions for library preparation. Whole-genome bisulfite sequencing was performed with the Ovation Ultralow Methyl-Seq System from NuGEN following the published procedure (http://www.nugen.com/products/ovation-ultralow-methyl-seq-library-systems) adapted from [75]. The resulting DNA library was purified and bisulfite converted. A qPCR assay was used to determine the optimal number of PCR amplification cycles (between 10 and 15 cycles) required to obtain a high diversity library with minimal duplicated reads prior to final library amplification. The sequencing was performed with paired ends (2 × 150 bp) on an Illumina HiSeq4000 platform. Raw data were stored in FASTQ files (SRA record is under the reference PRJNA635601, see Table 2).

### 3.3. Epigenomics Data Generation and Analysis

The bioinformatics pipeline used in this study is adapted from the ENCODE pipeline (https://www.encodeproject.org/wgbs/) and installed in a Galaxy framework (http://bioinfo.univ-perp.fr/, Perpignan, France). First, quality control and cleaning of the raw data were carried out by only considering nucleotides with a quality score over 26, and reads which had more than 95% of their nucleotides over this quality threshold. The second step was the alignment of the WGBS reads on the latest available genome sequence (https://doi.org/10.5061/dryad.08vv50n) [3] by using BISMARK (version 0.16.3, [77] and Bowtie 2 tools (version 2.1.0; [78]). Due to sequenced genome quality (2090 scaffolds in version 2, N residues in intergenic sequences), BSMAP version 2.74 [79,80] was used to measure DNA methylation levels at each cytosine site on a reduced representation consisting on gene models (34,363) including 5 Kbp flanking regions. All data were processed in R [81]. Multidimensional scaling was performed using only sites covered by at least seven reads were kept as recommended in the EdgeR differential methylation detection procedure [82]. Differential methylation in pairwise sample comparisons was detected following the DSS procedure [83]. It models the biological variation using neighboring C sites as pseudo-replicates and automatically adjusts library coverage differences. DMC were kept when the differential methylation *p*-value was below 0.05 (gene-wise Wald test on two-group comparisons). DMR were detected when at least 3 DMC were present in the same region of at least 50 bp. In addition, we summarized DMR at the gene levels by averaging methylation differences to give more robustness to the analysis [42] (Figure 4A and Figure 5).

### 3.4. Transcriptome Analysis

Gene expression was obtained from previously published RNA-seq data as well as new samples (flowers, basal, and distal parts of young leaves, deposited as project accession number PRJEB38854, see Table 2). RNA were obtained from plants as described above using a Trizol-based extraction [14]. Libraries prepared with the Illumina TruSeq RNA Library Prep Kit v2 were sequenced by Eurofins Genomics using the Illumina HiSeq2000/2500 technology, in the paired-end mode (2 × 100 pb). Sequencing runs yield a total of 25,352,619, 20,959,861, and 17,313,293 reads for samples ERR4235477, ERR4235478, and ERR4235479, respectively. Reads were first trimmed with FastP v0.20 [84] using default parameters and next pseudo-aligned on transcripts predicted from the second *C. roseus* genome version [3] and quantification was performed with Salmon v0.14 [85]. Differentially expressed genes were identified using the EdgeR R package [86], using a biological coefficient of variation of 0.2. Genes were considered as differentially expressed if the *p*-value of the exact test on the binomial fit were below 0.05.

### 3.5. Metabolite Analysis

Alkaloid content in samples depicted in Figure 5B was obtained from a meta-analysis of our previously published results. Briefly, flash-frozen samples were lyophilized and extracted in methanol. Strictosidine, tabersonine, lochnericine, vindoline, and catharanthine were quantified on the same cultivar (Apricot Sunstorm) in the diverse tissue types [6,10,14] using a LC-MS/MS apparatus. Identification was confirmed by retention times and mass spectra of authentic standards.

### 3.6. Statistical Procedures

All analyses were made within the R programming language [81]. Tables were all used in the data.table format [87]. All graphs were made with the ggplot2 package [88], together with the ggdendro [89] and ggrepel [90] extensions. The PCA in Figure 2 was constructed with the FactoMineR package [91]. Correlations in Figure 3C,D were calculated with the Pearson coefficient and tested with Pearson’s product moment correlation coefficient. Heatmaps were plotted with the pheatmap package [92]. Hierarchical trees were constructed with the Ward agglomeration criterion on Euclidean distances. Groups of DEM in Figure 4 were detected using a dynamic tree cutting algorithm [56] with a minimum cluster size of 30. Functional analysis of genes was performed by transferring annotations from Uniprot homologs (blastx, [93]) or using the eggNOG mapper [94]. Functional term (GO or Uniprot keywords) enrichment was tested using a hypergeometric distribution Appendix A). *P*-values were adjusted using the Bonferroni correction. The gene co-expression network (Figure 6) was constructed using a Spearman’s *rho*-based distance matrix and the igraph package [95]. Significant correlations were retained if *rho* > 0.95. When applicable, differences between treatments were calculated using the Student’s *t*-test or the non-parametric Wilcoxon rank sum test.

## 4. Conclusions

Our present study reports the first methylome using WGBS for the medicinal plant *C. roseus* with DNA methylation signatures specific to organs, ontogenesis, and environmental variations and their relationship with gene expression and MIA accumulation. Focusing on genes whose methylation state and expression levels varied significantly between samples, we found significant relationships depending on cytosine context and gene parts between DNA methylation and expression. In particular, variation of DNA methylation up- and downstream of genes as well as on CG (introns) or CHH (exons) sites may participate in modulating tissue-specific gene expression. This was clearly the case for the 12 functional detected gene groups involved in photosynthesis, stress response, hormone signaling, and specialized metabolism in agreement with the nature and characteristics of each tissue. This was also the case for the MIA pathway where 10 genes were found whose differential methylation and expression correlated with the tissue/organ/developmental/environmental specific MIA accumulations and their critical roles in MIA synthesis. These genes correspond to enzymes and to the three major transcription factors of the MIA pathways. In addition, 24 co-expressed transcription factors were identified and may participate in controlling the tissue-specificity of the MIA pathway. Therefore, DNA cytosine methylation seems to directly regulate expression of genes encoding enzymes and transcription factors, which in turn may indirectly promote or inhibit the expression of downstream MIA-related genes. While our integrative analysis supports a role of DNA methylation in the developmental and environmental control of specialized metabolisms in plants, other epigenetic modifications such as those targeting histones and non-coding RNAs deserve further investigations. The potential coordination between epigenetics and hormonal controls [96] for specialized metabolisms is also a future challenge and notably in the frame of improving the production of secondary metabolites for pharmaceutical applications using plant biotechnologies.

## Figures and Tables

**Figure 1 ijms-21-06028-f001:**
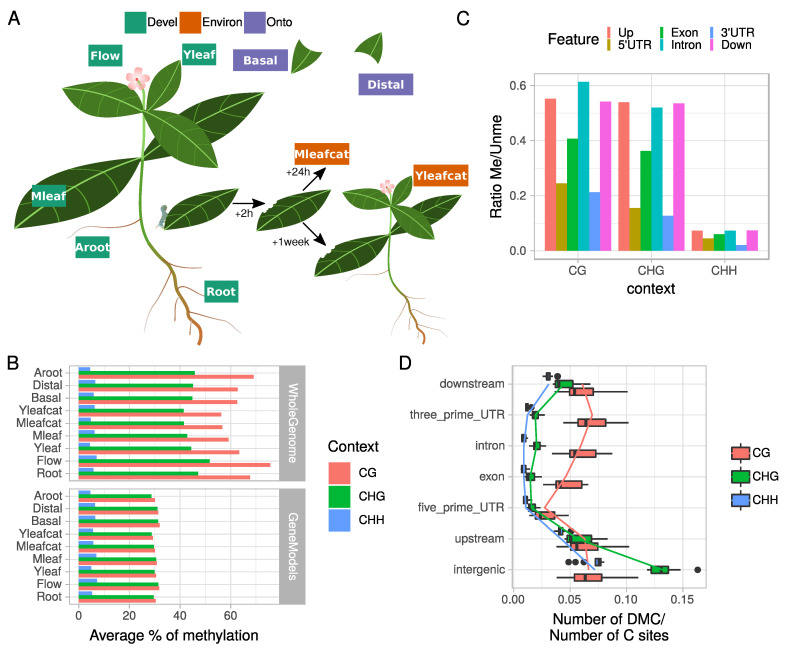
**Overview of the*****C. roseus*****experimental design for the WGBS and methylome data**. (**A**) Nine samples reflecting developmental variation within distinct organs (Leaf, Flower, Root or Adventitious Root) or during leaf ontogenesis (young versus mature; or basal versus distal) and environmental effects on leaf after direct (mature leaf 24 h post folivory) or indirect (distance young leaf 1 week post folivory) biotic attacks were chosen to characterize whole genome cytosine methylation profiles. Flow = Flower; Aroot = Adventitious root; Mleaf = Mature leaf; Yleaf = Young leaf; Basal = Basal part of leaf; Distal = Distal part of leaf; Mleafcat = Mature leaf post folivory by caterpillars; Yleafcat = Young leaf from subjected plants 1 week after folivory on mature leaves. (**B**) Average % of cytosine methylation in the whole genome or gene model targeted approaches. (**C**) Average cytosine methylation levels of different gene features in each cytosine context for the whole genome approach (**D**) ratio between the number of differentially methylated cytosines (DMC) and number of cytosine sites per feature and cytosine contexts (CG, CHG, or CHH). In the whole genome approach, DMCs were retained if the difference in methylation between the two samples considered was >25%.

**Figure 2 ijms-21-06028-f002:**
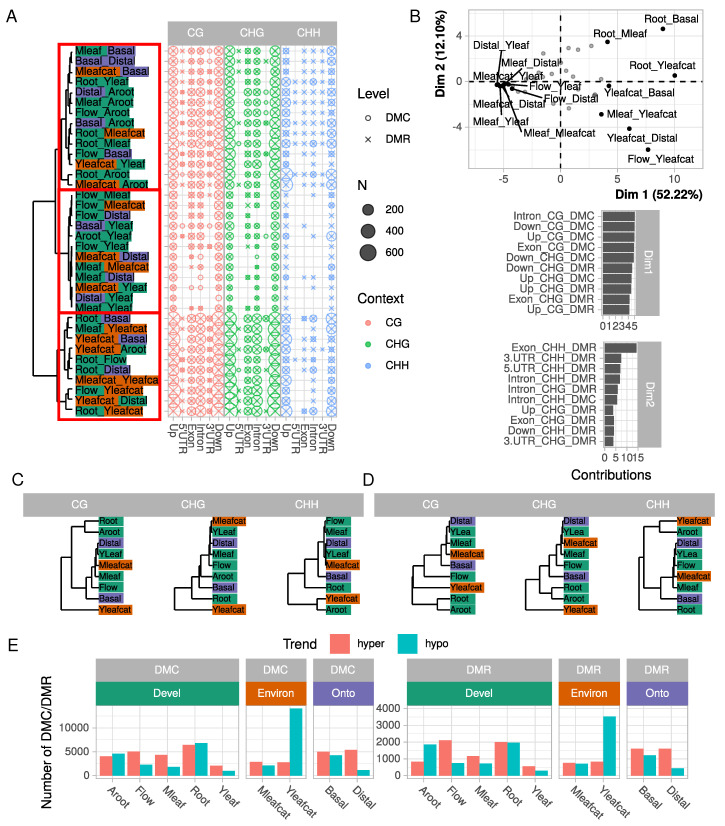
**Classification of samples according to DMC and DMR numbers and contexts**. (**A**) number of DMC and DMR in all 36 pairwise comparisons, for each context and gene model feature. Comparisons were clustered using an Euclidean distance calculated on the number of DMCs/DMRs and the Ward agglomeration criterion. (**B**) principal component analysis of DMC/DMR number. Comparisons are projected on the two first dimensions but names are indicated only for the 15 highest contributions. For both dimensions, the five highest variable contributions are represented below. (**C**,**D**) samples clustered according to the numbers of DMC and DMR, respectively. Euclidean distances were calculated on square matrices containing samples in rows and columns. Clustering was constructed according to Ward agglomeration criterion; (**E**) number of hyper- and hypo-methylated DMCs/DMRs for each sample compared with all eight others.

**Figure 3 ijms-21-06028-f003:**
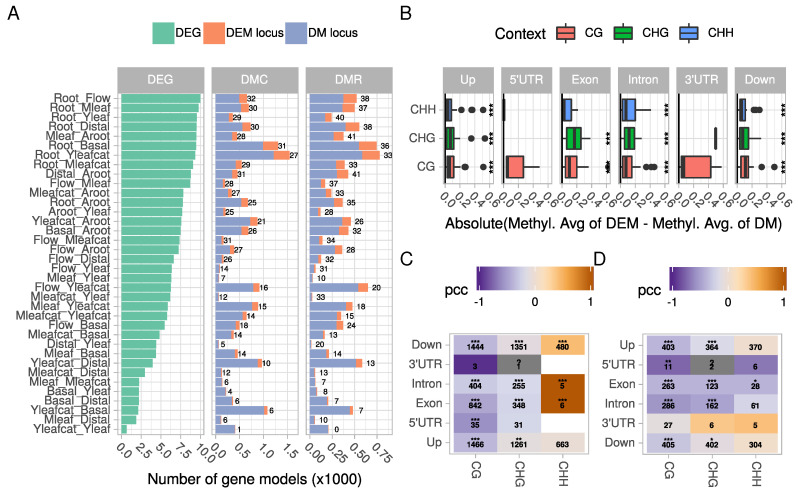
**Relationship between differential methylation and differential expression.** (**A**) number of differentially expressed genes (DEG), and number of gene models harboring at least one DMC or one DMR. A DM (differentially methylated) locus corresponds to a non-DEG with at least one DMC or DMR, while a DEM (differentially expressed and methylated) locus corresponds to a DEG with at least one DMC or DMR. Numbers correspond to the % of DEM/DM. Please note the differences in scale on the *x*-axis; (**B**) methylation differences of DEM or DM genes were averaged per comparison, context and gene feature. Differences between averages were calculated to highlight variation in methylation profiles between DEM and DM genes. The boxplots summarize data from all comparisons using absolute values of average differences. Asterisks denote statistical significant differences (Wilcoxon rank sum test, Bonferroni corrected *p*-values, ***: *p* < 0.001). (**C**,**D**) Pearson correlation coefficients (PCC) measuring the correlation between a gene expression log2FC and its methylation state (“.” *p* < 0.1, “*” *p* < 0.05, “**” *p* < 0.01, “***” *p* < 0.001) at the DMC and DMR levels, respectively. Numbers in cells correspond to the number of points used to calculate correlations.

**Figure 4 ijms-21-06028-f004:**
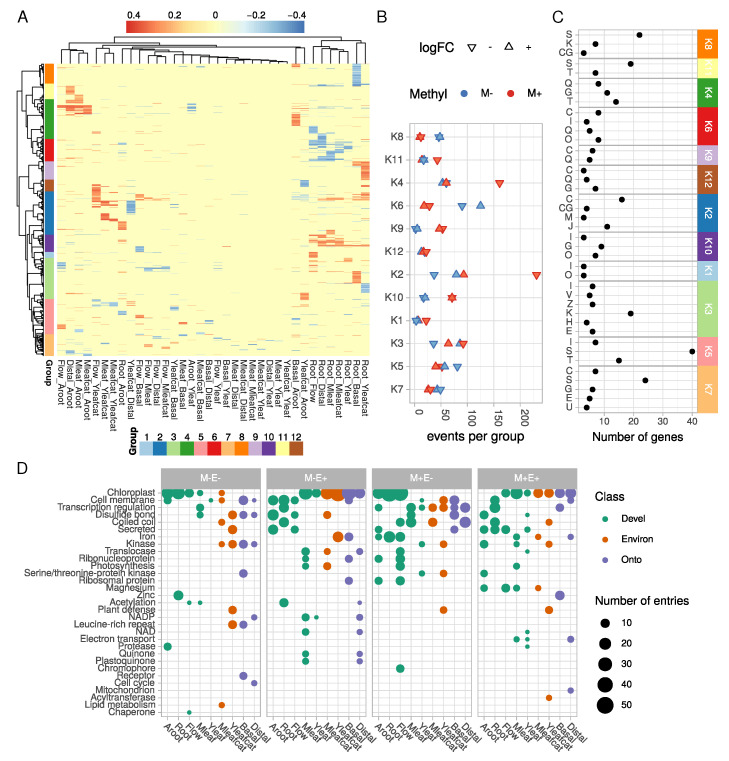
**Characterization of differentially expressed and methylated (DEM) genes**. (**A**) heatmap of DNA methylation differences for each differentially expressed and methylated gene model and each comparison. Cell color is proportional to the methylation difference (blue to red scale). Comparisons and gene loci are clustered according to the Euclidean distance and the Ward’s aggregation criterion. (**B**) for each group in (**A**) DEM events (one gene in one comparison) were counted for each trend (over/under expressed (logFC) and hypo/hyper methylated (Methyl)). (**C**) EggNOG functional annotation of gene clusters from the heatmap in A. Only enriched classes are shown (hypergeometric test, Bonferroni ajdusted-*p*-value < 0.05). A, RNA processing and modification; B, Chromatin structure and dynamics; C, Energy production and conversion; D, Cell cycle control, cell division, chromosome partitioning; E, Amino acid transport and metabolism; F, Nucleotide transport and metabolism; G, Carbohydrate transport and metabolism; H, Coenzyme transport and metabolism; I, Lipid transport and metabolism; J, Translation, ribosomal structure and biogenesis; K, Transcription; L, Replication, recombination and repair; M, Cell wall/membrane/envelope biogenesis; N, Cell motility; O, Post-translational modification, protein turnover, and chaperones; P, Inorganic ion transport and metabolism; Q, Secondary metabolites biosynthesis, transport, and catabolism; R, General function prediction only; S, Function unknown; T, Signal transduction mechanisms; U, Intracellular trafficking, secretion, and vesicular transport; V, Defense mechanisms; W, Extracellular structures; Y, Nuclear structure; Z, Cytoskeleton. (**D**) most represented Uniprot keywords for DEM genes in for each sample and methylation/expression profile. For each sample, all nine comparisons were made to identify gene sets following one the four profiles: M-E-, hypomethylated and decreased expression; M-E+, hypomethylated and overexpressed; M+E-, hypermethylated and decreased expression; M+E+, hypermethylated and overexpressed. Each corresponding Uniprot entry was mapped to Uniprot keywords, which were then summarized for each sample.

**Figure 5 ijms-21-06028-f005:**
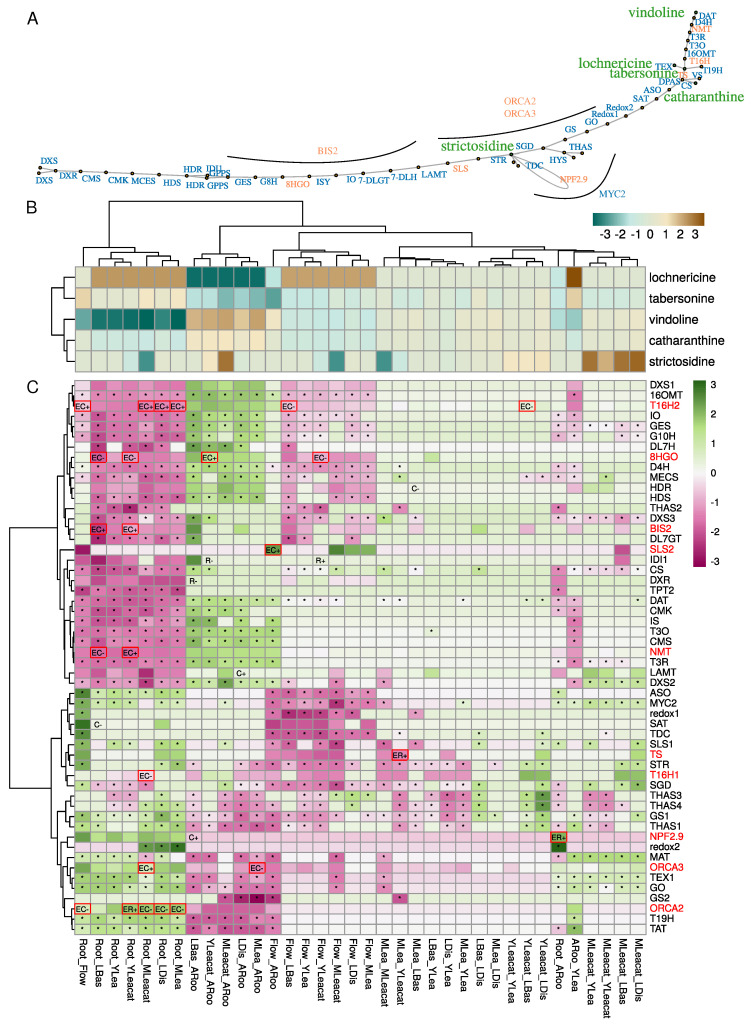
**Methylation and expression variation in the MIA pathway.** (**A**) MIA pathway. Differentially methylated genes are indicated in orange. Transcription factors BIS, ORCA2/3, and MYC2 are positioned close to their known targets. (**B**) Analysis of differential accumulation of metabolites. Each cell in the table represents the log10 transformed ratios of metabolite between samples. (**C**) differential expression analysis of genes involved in the MIA pathway. Cells show scaled log2 fold changes. *, differentially expressed (Sample1_Sample2). C−, hypomethylated DMC in sample1; C+, hypermethylated DMC in sample1. R-, hypomethylated DMR in sample1; R+ hypermethylated DMR in sample1. When a “E” is indicated before, it indicates that the gene is both differentially methylated and expressed. Differentially expressed and methylated genes (EC or ER) are surrounded by red boxes and their names indicated in red.

**Figure 6 ijms-21-06028-f006:**
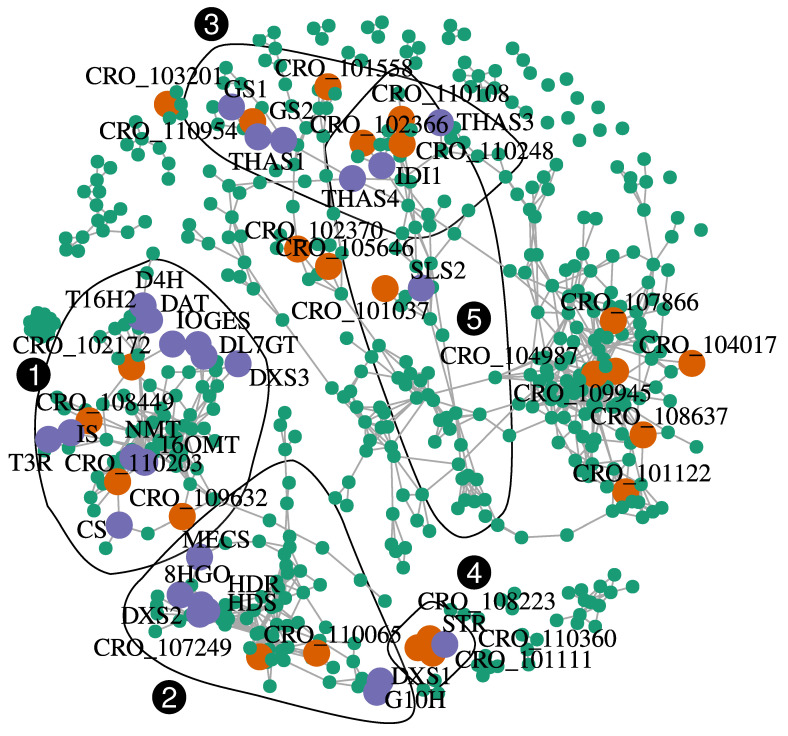
**Co-expression network (Spearman’s rho correlation-based) visualizing transcriptional relationships among MIA genes and other differentially methylated and expressed genes detected in this study**. MIA genes are indicated by purple points and black labels, while differentially expressed and methylated genes are indicated in turquoise or orange if they code potential transcription factors (Table 1). Genes are connected by grey segments if their Spearman’s rho coefficient was above 0.95. Black polygons correspond to communities detected with a fast greedy modularity optimization based algorithm. Only communities containing at least one MIA gene are shown. Numbers in white in black circles correspond to community identification numbers.

**Table 1 ijms-21-06028-t001:** **Description of differentially expressed and methylated transcription factors co-expressed with MIA genes.** Expression LogFC and methylation differences are given for the most contrasting comparison.

Locus Tag	Best Hit in Uniprot Database (blastx)	% id	e-Value	Comparison	LogFC	Methylation Difference
**Community 1 (GES, IS, IO, and DL7GT, T16H2, 16OMT, T3R, NMT, D4H, and DAT, vindoline-related community)**						
CRO_T102172	DOF56_ARATH, Dof zinc finger protein DOF5.6 (AtDOF5.6)	71.77	4 $\times \;10^{-43}$	Distal vs. Aroot	2.54	0.31
CRO_T109632	BH048_ARATH, Transcription factor bHLH48 (AtbHLH48)	40.55	3 $\times \;10^{-49}$	Root vs. Basal	−2.27	0.29
CRO_T110203	WOX1_ARATH, WUSCHEL-related homeobox 1	56.95	4 $\times \;10^{-43}$	Root vs. Yleafcat	−9.16	0.22
**Community 2 (DXS1, DXS2, HDS, HDR, MECS, G10H, and 8HGO; terpene precursor community)**						
CRO_T107249	UNE10_ARATH, Transcription factor UNE10 (AtbHLH16) (bHLH 16)	52.71	5 $\times \;10^{-25}$	Root vs. Mleaf	−8.76	0.19
CRO_T107392	MYB06_ANTMA, Myb-related protein 306	61.43	3 $\times \;10^{-85}$	Root vs. Flower	−6.72	−0.29
CRO_T110065	UNE10_ARATH, Transcription factor UNE10 (AtbHLH16) (bHLH 16)	46.13	2 $\times \;10^{-51}$	Yleafcat vs. Aroot	2.44	0.24
**Community 3 (GS1, GS2, THAS1, THAS3, and THAS4, strictosidine aglycone acting enzymes)**						
CRO_T110954	BH113_ARATH, Transcription factor bHLH113 (AtbHLH113)	42.74	1 $\times \;10^{-39}$	Yleafcat vs. Aroot	−1.85	0.18
**Community 4 (ORCA3 and STR; control of strictosidine accumulation)**						
CRO_T101111	BH030_ARATH, Transcription factor bHLH30 (AtbHLH30)	40.18	5 $\times \;10^{-30}$	Root vs. Basal	2.92	−0.28
CRO_T108223	PTI6_SOLLC, Pathogenesis-related genes transcriptional activator PTI6	59.51	3 $\times \;10^{-39}$	Root vs. Yleafcat	5.66	0.26
**Community 5 (SLS2 and IDI1)**						
CRO_T101037	PCF5_ORYSJ, Transcription factor PCF5	85.92	4 $\times \;10^{-35}$	Flower vs. Yleafcat	3.72	−0.29
CRO_T102366	BH079_ARATH, Transcription factor bHLH79 (AtbHLH79) (bHLH 79)	57.19	2 $\times \;10^{-80}$	Flower vs. Aroot	5.46	−0.24
CRO_T110108	YAB1_ARATH, Axial regulator YABBY 1 (Fl-54)	65.12	3 $\times \;10^{-82}$	Mleafcat vs. Yleafcat	−2.41	−0.32
CRO_T110248	ICE1_ARATH, Transcription factor ICE1 (AtbHLH116)	60.62	2 $\times \;10^{-102}$	Flower vs. Mleafcat	1.74	−0.26

**Table 2 ijms-21-06028-t002:** **WGBS** **and** **RNA-seq data.**

Sample	Accession	Reference	Project Accession Number
**Root** WGBS RNA-seq	SRR11932631 SRR1271858	This study [76]	PRJNA635601 PRJNA246273
**Adventitious root** WGBS RNA-seq	SRR11932632 ERR2112587	This study [10]	PRJNA635601 PRJEB22378
**Young Leaf** WGBS RNA-seq	SRR11932629 ERR1512377	This study [14]	PRJNA635601 PRJEB14626
**Mature Leaf** WGBS RNA-seq	SRR11932637 ERR1512375	This study [14]	PRJNA635601 PRJEB14626
**Mature leaf 24 h post folivory** WGBS RNA-seq	SRR11932636 ERR1512373	This study [14]	PRJNA635601 PRJEB14626
**Young leaf from folivory subjected plants** WGBS RNA-seq	SRR11932635 ERR1512376	This study [14]	PRJNA635601 PRJEB14626
**Flower** WGBS RNA-seq	SRR11932630 ERR4235477	This study This study	PRJNA635601 PRJEB38854
**Young Leaf basal part** WGBS RNA-seq	SRR11932634 ERR4235478	This study This study	PRJNA635601 PRJEB38854
**Young Leaf distal part** WGBS RNA-seq	SRR11932633 ERR4235479	This study This study	PRJNA635601 PRJEB38854

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
