# Peer review of "Developmental Methylome of the Medicinal Plant Catharanthus roseus Unravels the Tissue-Specific Control of the Monoterpene Indole Alkaloid Pathway by DNA Methylation"

_ijms, 2020, doi:10.3390/ijms21176028_

Round 1

Reviewer 1 Report

Dear Authors,

Generally, I rate the value of the manuscript highly, although I have a few comments on it.

Editorial comments:
The first page is missing a headline with the name and logo of the journal.
When using Latin names for the first time, they should be full with the author's name and always written in italics. In many places it is different. (lines: 20, 69, 82, 120, 132, 269, 306).
In line 141 B. distachyon and in line 242 A. thaliana instead of the full name.
When using the common name of a species, please give the full Latin name in parentheses.
Line 68: cis and trans should be italic.
Line 313 vs should be italic because it abbreviates the Latin word versus.
Figure 2a: The lettering on the dendrogram should be black font as it is on 2c and 2d - now it is illegible.
Figures 2b and 4a must be larger - now they are illegible.

The description of the methodology lacks relevant information:
Line 441 please specify the manufacturer of Trizol reagent.
How was the molarity of the samples assessed?
What kit was used for HiSeq analysis?
Methodology for RNAseq is missing. Which reagents? Which platform? How many readings per sample? What length of reading?

My major concern relates to the lack of confirmation of the expression of the most important genes using qPCR. Please add the results of such analysis, or explain thoroughly why it is missing.

Regards,

Reviewer

Reviewer 2 Report

Dear authors,

The mansucript wa written flluently and informative and was easy to follow and at the same time the content was so intersting to read.

I just have minor suggestions/ questions:

1- Lines 33 and 61, 130, 146, 317-318: I am just wondering why TABERSONINE and CATHARANTHINE; PHYTOCHROME INTERACTING FACTORS SYNTHASE; Up and Downstream; Upstream, CROCETIN GLUCOSYLTRANSFERASE (CRO_T108522), 7-DEOXYLOGANETIN GLUCOSYLTRANSFERASE LIKE (CRO_T106923) and several other gene names are in capital letters.

2- Should not authors open bHLH as basic helix-loop-helix in the first time it appear?

3- Line 158: should Yleafcat (and other abbreviations) be explained for the first time in the text, though there are list of abbreviations in the Figure 1.

4- These articles could be good addition to your discussion.

https://doi.org/10.5897/JMPR2019.6803

Plant Biotechnol J. 2019 Apr; 17(4): 826–835.

Hortic Res. 2020; 7: 112.

https://doi.org/10.1371/journal.pone.0191492.

Good Luck!

Round 2

Reviewer 1 Report

The manuscript should be accepted in its current form.